# In Vitro Evaluation of the Combinatorial Effect of Naringenin and Miltefosine against *Leishmania amazonensis*

**DOI:** 10.3390/ph17081014

**Published:** 2024-08-01

**Authors:** Vinícius Lopes Lessa, Gustavo Gonçalves, Beatriz Santos, Victoria Cruz Cavalari, Rafael Felipe da Costa Vieira, Fabiano Borges Figueiredo

**Affiliations:** 1Carlos Chagas Institute, Oswaldo Cruz Foundation (Fiocruz), Curitiba 81310-020, PR, Brazil; ggustavogonsalves@gmail.com (G.G.); beatrizsantos.biobs@gmail.com (B.S.); viccavalari97@gmail.com (V.C.C.); fabiano.figueiredo@fiocruz.br (F.B.F.); 2Graduate Program in Veterinary Sciences, Federal University of Paraná, Curitiba 80035-060, PR, Brazil; 3Department of Epidemiology and Community Health, College of Health and Human Services, University of North Carolina at Charlotte, Charlotte, NC 28223, USA; rvieira@charlotte.edu; 4Center for Computational Intelligence to Predict Health and Environmental Risks (CIPHER), University of North Carolina at Charlotte, Charlotte, NC 28223, USA

**Keywords:** flavonoids, naringenin, n-hexadecylphosphonocholine, leishmaniasis, cutaneous leishmaniasis

## Abstract

*Leishmania amazonensis* causes a clinical form called diffuse cutaneous leishmaniasis (DCL) with challenges to treatment, like low efficiency and drug toxicity. Therefore, it is necessary to investigate new therapies using less toxic leishmanicidal compounds, such as flavonoids like naringenin, and their combination with conventional drugs, such as miltefosine. Antileishmanial dose/response activity, isobologram, calculation of dose reduction index (DRI), and fractional inhibitory concentration index (FICI) tests were performed on in vitro assays using reference promastigote forms of *L. amazonensis* (IFLA/BR/67/PH8) to assess the combinatorial effect between naringenin and miltefosine. The in vitro results of isobologram, DRI, and FICI calculations showed that the combination of the compounds had an additive effect and was able to reduce the half maximal inhibitory concentration (IC_50_) of miltefosine in the promastigote forms of the parasite compared to the treatment of the drug alone. This study demonstrated in vitro the viability of a combination action of the flavonoid with the treatment with miltefosine, opening space for further investigations on the association of natural compounds with the drugs used for the treatment of *L. amazonensis*.

## 1. Introduction

Leishmaniasis is a disease caused by single-celled protozoan of the genus *Leishmania*. The three main forms of the disease are visceral leishmaniasis (VL), mucocutaneous leishmaniasis (CML), and cutaneous leishmaniasis (CL). In the Americas, 18 countries are endemic to CL and CML, with Brazil, Colombia, Peru, Nicaragua, and Bolivia having the highest estimated case counts [1]. Among the species that cause CL in Brazil, *Leishmania amazonensis* has an incidence in primary and secondary forest areas of the legal Amazon (Amazonas, Pará, Rondônia, Tocantins, and Maranhão) and also in the northeastern (Bahia), southeastern (Minas Gerais and São Paulo), central–western (Goiás), and southern (Paraná) states. This species causes a wide spectrum of clinical forms, which are localized cutaneous leishmaniasis (LCL), CML, diffuse cutaneous leishmaniasis (DCL), and disseminated leishmaniasis (DL) [2].

Although DCL is relatively rare, it is extremely severe; diffuse skin infiltration and a large number of nodules and papules clinically characterize this form. Patients affected by DCL present lesions that cover the entire body, predominantly in the extremities, which rarely involve the nasopharyngeal mucous membranes [2]. Among the clinical forms caused by *L. amazonensis*, DCL is difficult to treat with conventional drugs; this clinical form has as its main characteristics the Th2 response with expression of interleukin-10, IL-4, and low expression of interferon gamma (IFN), leading to the patient’s anergy to the parasite, high parasitic loads, and deforming lesions that generate physical stigmas and psychological impacts [3]. In addition to the ineffective immune response to DCL, the use of meglumine antimoniate and second-choice drugs like liposomal amphotericin B and pentamidine isethionate proved unsuccessful in cases of refractoriness [3]. Treatments with first- and second-choice drugs in the different clinical forms caused by *L. amazonensis* present some complications. These complications include toxic effects and restricted access of the population with economic vulnerability to medical care and follow-up, especially in the northern and northeastern Brazilian regions, due to the lack of resources and professionals. Additionally, drug resistance to treatment by meglumine antimoniate is an aggravating factor [4]. Miltefosine (n-hexadecylphosphonocholine), a phospholipid that is the hexdecyl monoester of phosphocholine with a molecular weight of 407.6 g/mol, shows a lower toxicity when compared with the meglumine antimoniate, easy oral administration, and an efficiency rate in the elimination of the parasite close to that of first- and second-choice drugs [5,6].

However, due to their teratogenic effect, new therapeutic alternatives demonstrating similar and less toxic efficacy have been explored. Efforts have been directed toward treatment based on plants and their metabolites, in particular the flavonoids, which have properties as preventive agents against cancer, antioxidant activity, and leishmanicidal activity [7,8]. Herein, we selected the flavonoid naringenin (5,7-Dihydroxy-2-(4-hydroxyphenyl) chroman-4-one), which is a trihydroxyflavanone, a flavanone substituted by hydroxyl groups at positions 5, 6, and 4 with molecular weight of 272.25 g/mol. In addition to its important immunomodulatory properties [5,9], naringerin has demonstrated potent in vitro activity against promastigotes and amastigotes of *Leishmania donovani* [10]. Considering the effects of naringenin and the fact that miltefosine is the drug with less toxic effects, we sought to associate these two compounds and investigate their therapeutic potential against *L. amazonensis*.

## 2. Results

### 2.1. In Silico Study

The physicochemical properties of naringenin and miltefosine were assessed to compare their predicted oral bioavailability using Lipinski’s and Veber’s criteria. Miltefosine presents one violation and naringenin none (Table 1). The structures of miltefosine and naringenin are represented in Figure 1a and Figure 1b, respectively; for both substances, pan assay interference compounds (PAINS) were not identified by the SwissADME web tool.

### 2.2. Growth Curve

The growth of parasites presented an adaptation phase up to 48 h; after this period, the parasites reached their exponential growth phase in up to 96 h, reaching the stationary phase following 120 h of cultivation.

### 2.3. Antipromastigote Activity In Vitro

The mean IC_50_ obtained from the treatment with miltefosine alone was 13.20 µM, while naringenin had an IC_50_ of 219.86 µM. For the synergism tests, the IC_50_ values of the proportions 1:1, 2:1, 4:1, and 6:1 were first obtained, allowing the calculation of ΣFICI (Table 2) and enabling the construction of the isobologram (Figure 2). Calculating the average of the sum of FICI, we arrived at the value of χΣFICI, which was 0.803. It was observed that the interaction between the compounds is additive in all proportions tested.

## 3. Discussion

The compound combination strategy is adopted to optimize treatments for a wide range of diseases. Past studies have reported treatment regimens where antibiotics are combined to combat multidrug-resistant bacteria like *Acinetobacter baumannii*, reducing the mortality of infected patients [12], cancer cells such as breast cancer [13], and different types of viruses like HIV, HCV, and influenza [14]. The combination of compounds for treating leishmaniasis is also adopted to improve the effectiveness of treatments, diminishing their cost and time and reducing the likelihood of the emergence of resistant parasite strains. Compared to monotherapies, combined drug therapies present greater stimulation in the activity of leukocytes and their replication during infection, as well as increased production of cytokines that regulate the Th1 and ROS responses acting on the elimination of the parasite [15]. Faced with the challenges of monotherapies with first- and second-choice drugs, known by their side effects that hinder treatment, some efforts have been made to search for antiparasitic compounds in plants. In this sense, flavonoids stand out, which consist of a large group of phenolic compounds synthesized by the phenylpropanoid pathway in plants and have several antimicrobial, anti-cancer, and leishmanicidal activities [10,16]. In addition to treatments with isolated flavonoids, a previous study evaluated the optimization of the miltefosine treatment. When the drug was associated with the flavonoid apigenin, a reduction in parasitemia in mice was obtained with only half the dose of the drug when compared with the treatment of the drug alone [17]. As observed in the in silico analysis using the SwissADME web tool, naringenin presents physicochemical properties that favor its bioavailability, opening space for further investigation of its leishmanicidal properties in isolation and in combination. We investigated the combinatorial effect of miltefosine with naringenin through using synergism and isobolagram testing to evaluate the interaction between the natural compound and the commercial drug to optimize the treatment of the drug, reducing its toxicity by associating it with a non-toxic natural compound [18]. The MTT assay performed herein showed values of the IC_50_ of the different proportions (Table 2). A reduction in miltefosine IC_50_ values was observed in all proportions; it was possible to assess that in the 1:1 and 2:1 proportions there was a substantial reduction in the IC_50_ dose, and to evaluate this reduction, the DRI calculation was used [19]. The DRI calculation measures how many times the dose of each drug in combination can be reduced due to the level of interaction up to a certain level of effect when compared to separate treatment. A DRI = 1 value indicates that there is no dose reduction; if the DRI > 1, it indicates a favorable dose reduction that leads to toxicity reduction; if the DRI < 1, the dose of the drug is not favorable for reduction. That is, it can be observed that in the proportions of 1:1, 2:1, and 4:1, they provided a reducing dose of miltefosine without reducing its effect, which, on the contrary, had a greater effect at lower doses than in the trial with the drug alone. The calculation of ΣFICI made it possible to construct the isobologram for better visualization of the interaction between the compounds. In Figure 2, it can be seen that the proportions of 1:1, 2:1, and 4:1 were below the indifference line with ΣFICI values of 0.53, 0.57, and 0.96, respectively. The 6:1 ratio presented a ΣFICI value of 1.16, slightly above the indifference line. According to the criteria for evaluating the interaction between compounds described in Section 4.3, it was concluded that the compounds in all proportions showed additive behavior (1 > ΣFICI > 0.5), with the ΣFICI values and the χΣFICI value of 0.803, defining the interaction between the compounds in general as additive. The additive effect between two drugs commonly refers to non-interaction or inertism between the substances being observed when the effect of the combination between different drugs is the sum of the effects of these drugs tested separately [19]. In general, it establishes a demarcation between the synergistic and antagonistic natures in the investigation of the interaction between drugs. Additive interactions have already been observed in several studies with combinations of leishmanicidal compounds [19,20,21]. The use of drugs with an additive effect in a combination can reduce chances of resistance, bringing the possibility of shortening treatment time [20], making this strategy attractive to mitigate the adverse effects present in the treatment of leishmaniasis. In summary, the combination of naringenin with miltefosine in vitro was able to potentiate the action of the drug, reducing the IC_50_ of the drug by approximately two times and requiring a lower dose of miltefosine to eliminate a considerable percentage of *L. amazonensis*.

## 4. Materials and Methods

### 4.1. In Silico Study

The structures of naringenin and miltefosine were used to evaluate their theoretical physicochemical properties and the presence of PAIN. The predictions were calculated using the SwissADME web tool [22], considering Lipinski’s rule of five (RO5) [23], followed by the additional rule proposed by Veber [24].

### 4.2. Growth Curve

Promastigote forms of the *L. amazonensis* strain (IFLA/BR/67/PH8) were maintained in an M199 medium with Hanks’ salts (Sigma-Aldrich Brazil Ltd., São Paulo, Brazil) at 25 °C for inoculum production to be used in our experiments. The culture was maintained by replication every three to four days. The number of cells was measured in a hemocytometer and optical microscope, and the growth curve was performed in triplicate.

### 4.3. Antipromastigote Activity In Vitro

The in vitro antipromastigote activity of miltefosine and naringenin was evaluated in promastigote forms of *L. amazonensis* using the 3-[4,5-dimethyl-thiazol-2-yl]-2,5-diphenyl-tetrazolium bromide (MTT) (Sigma-Aldrich Brazil Ltd., São Paulo, Brazil) assay, with the drug tested alone and combined with naringenin [25] for a 48-h treatment. Commercial naringenin with a purity of 98% (Sigma-Aldrich Brazil Ltd., São Paulo, Brazil) and commercial miltefosine (Cayman Chemical, São Paulo, Brazil) were used, and stock solutions with concentrations of 50 mg/mL and 1 mg/mL were prepared in DMSO (Sigma-Aldrich Brazil Ltd., São Paulo, Brazil). We performed these tests to construct an isobologram to obtain the values of the FICI. The parasites were incubated at 25 °C for 48 h of treatment. After incubation, 50 µL of MTT solution (10 mg/mL) was added to each well. The plates were maintained at 37 °C for 4 h. To solubilize the formazan crystals, 20 µL of 10% sodium dodecyl sulfate (SDS) (Sigma-Aldrich Brazil Ltd., São Paulo, Brazil) and 50 µL of 100% DMSO were added to each well. Plate readings were performed on a microplate reader using a wavelength of 550 nm. All tests were performed in biological triplicate and technical quintuplicate. The results are expressed as the compound concentration capable of inhibiting parasite growth by 50% (IC_50_). The assay was performed in a 96-well flat-bottom microplate with a final volume of 200 µL. To perform the assay, the concentration of promastigote forms kept in the exponential phase was adjusted to 1 × 10^6^ cells/mL. To calculate the IC_50_ of miltefosine alone, seven dilutions of the initial 200 µM solution of the drug were performed, where each new 2× dilution of the initial dose, the ranges of 100 µM, 50 µM, 25 µM, 12.5 µM, 6.25 µM, 3.125 µM, and 1.5625 µM were obtained in the plate. In addition, an assay was performed with the isolated flavonoid. Following the same dilution strategy used for miltefosine but with different ranges of concentrations that are 1376.4 µM, 688.2 µM, 344.1 µM, 172.05 µM, 86.025 µM, 43.015 µM, 21.50 µM, and 10.75 µM, for these tests, the parasites were first seeded in the plate with M199 medium containing 0.75% DMSO with naringenin. After carrying out tests to evaluate the antipromastigote activity of the compounds in isolation, the next step was to carry out tests combining doses of naringenin and miltefosine in proportions of 1:1, 2:1, 4:1, and 6:1. With the IC_50_ values of the proportions, it was possible to evaluate the interactions of naringenin and miltefosine in the parasite’s growth by calculating the fractional inhibitory concentration index (FICI) [26] and the dose reduction index (DRI) to evaluate the possibility of a decrease in the dosage of miltefosine in the combination assay without reducing its effect [19]. The isobologram was plotted with the sum of the FICIs and the averages of the sums of the FICI ratios, with ΣFICI calculated to determine the nature of the interaction between the natural compound and the drug, with ΣFICI ≥ 0.5 indicating a synergistic effect, 0.5 >ΣFICI < 4 additive effect, and ΣFICI > 4 antagonistic effect [27]. The formulas used to calculate the FICI and DRI are described below:(1)$$\Sigma FICI=\left[\frac{{IC}_{50}drug\;A\;combination}{{IC}_{50}drug\;A\;alone}\right]+\left[\frac{{IC}_{50}drug\;B\;combination}{{IC}_{50}\;drug\;B\;alone}\right]$$
(2)$$DRI=\left(\frac{{IC}_{50}drug\;alone}{{IC}_{50}drug\;combination}\right)$$

### 4.4. Statistical Analysis

Statistical analysis and the construction of the isobologram was performed using GraphPad Prism 8.0.1 (GraphPad Software, Inc., San Diego, CA, USA). The results of the IC_50_ assays were transformed into log values and analyzed by a dose–response inhibition curve. A *p*-value ≤ 0.05 was considered statistically significant. The results were plotted with their means and standard deviations. The experiments were realized in biological triplicates and technical quintuplicates.

## 5. Conclusions

Although further work is needed, including tests with amastigotes, these results show that the association of naringenin with miltefosine has promising antileishmanial activity in vitro experiments, demonstrating that the flavonoid alone has an antipromastigote effect in *L. amazonensis* and can optimize the treatment of the commercial drug, increasing its efficiency.

## Figures and Tables

**Figure 1 pharmaceuticals-17-01014-f001:**
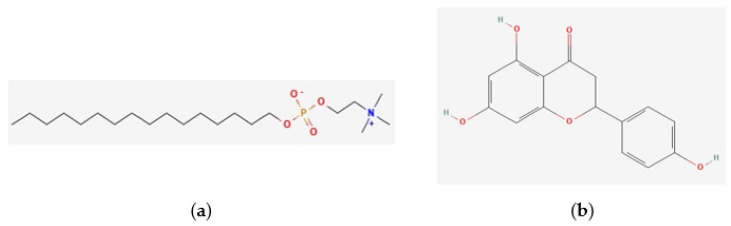
The structures of miltefosine and naringenin. (**a**) Miltefosine presents a molecular formula of C_21_H_46_NO_4_P (PubChem CID: 3599). (**b**) Naringenin presents a molecular formula of C_15_H_12_O_5_ (PubChem CID: 932). These structures were generated by PubChem [11].

**Figure 2 pharmaceuticals-17-01014-f002:**
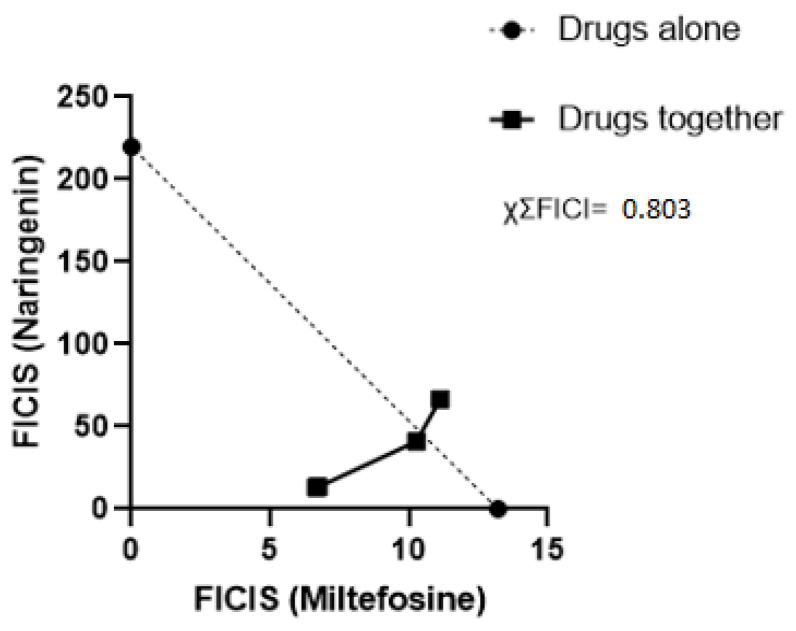
Isobologram analyzing the interaction of miltefosine and naringenin. Each point represents the FICI averages of the proportions 1:1, 2:1, 4:1, and 6:1 for naringenin and miltefosine; the points of the proportions 1:1 and 2:2 are overlapped due to the proximity of their FICI values. The dotted line represents the line of indifference. The result of χΣFICI is located at the top right.

**Table 1 pharmaceuticals-17-01014-t001:** Molecular properties of naringenin and miltefosine according to Lipinski’s and Veber’s criteria and the number of pan assay interference compounds according to SwissADME. MW: molecular weight; Log Po/w: Log of partition coefficient (consensus LogP on SwissADME); RB: number of rotable bonds; H-Acc: number of hydrogen bond acceptors; H-Don: number of hydrogen bond donors; tPSA (Å^2^): molecular polar surface area; PAINS: pan assay interference compounds.

	Naringenin	Miltefosine	Limit
MW	272.25 g/mol	407.57 g/mol	≤500
Log Po/w	1.84	3.35	≤5
RB	1	20	≤10
H-Acc	5	4	≤10
H-Don	3	0	≤5
tPSA (Å^2^)	86.99 Å^2^	68.4 Å^2^	≤140
PAINS	0	0	-
Violations	0	1	-

**Table 2 pharmaceuticals-17-01014-t002:** Representation of the IC_50_ values of miltefosine and naringenin in different proportions. FICI and DRI values were calculated using the formulas described in Section 4.3 of the Materials and Methods section. The IC_50_ values are represented with their mean and standard deviation; the FICI values are represented by the means extracted from the IC_50_ values. The DRI values are represented by means obtained by the calculation of the IC_50_ means of miltefosine in treatment alone and in combination with naringenin.

Proportion	IC_50_ Miltefosine (µM)	IC_50_ Naringenin (µM)	FICI	DRI
01:01	6.65 ± 0.9543	6.657 ± 0.9543	0.53	1.984277
02:01	6.73 ± 3.512	13.468 ± 7.029	0.57	1.961879
04:01	10.27 ± 3.832	41.08 ± 15.32	0.96	1.285955
06:01	11.12 ± 3.834	66.77 ± 23.00	1.14	1.186965
00:00	13.21 ± 3.125	219.9 ± 12.46		

## Data Availability

The data presented in this study are available upon request from the corresponding author.

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
