# Peer review of "In Vitro Evaluation of the Combinatorial Effect of Naringenin and Miltefosine against Leishmania amazonensis"

_pharmaceuticals, 2024, doi:10.3390/ph17081014_

Round 1

Reviewer 1 Report

Comments and Suggestions for Authors

The manuscript entitled '' In vitro evaluation of the combinatorial effect of naringenin and miltefosine against Leishmania amazonensis '' is not good from the medical viewpoint and is not well described.

·       Please, highlight and emphasize the importance and novelty of the study

·       The logic of the manuscript and introduction is not clear. The authors also should cite more recent and relevant references to cover the relevant literature.

·       The resolution of some Figures is unsatisfactory.

·       I also want to confirm whether these structures were been drawn in the program or from internet sources.

·       There is no in vivo study to support your work.

·       Authors need to correct some grammatical mistakes.

·       Please check the abbreviations throughout the manuscript.

·       The discussion needs more improvement.

·       The source of naringenin and miltefosine is not clarified.

·       The authors should cite more recent and updated references

Comments on the Quality of English Language

·       Authors need to correct some grammatical mistakes.

Author Response

 We appreciate your considerations. The answers of your commentaries are described below.

  1. Please, highlight and emphasize the importance and novelty of the study.

Answer: The novelty of the study were highlight and emphasized in the manuscript. Thanks for your suggestion.

  1. The logic of the manuscript and introduction is not clear. The authors also should cite more recent and relevant references to cover the relevant literature.

Answer: Thank you for your comment, we improved the text and references to format of a brief report to accomplish a recommendation of others rewires too.

  1. The resolution of some Figures is unsatisfactory

Answer:  Thank you for this recommendation; we improved the resolution of the figures in the manuscript.

  1. ·   I also want to confirm whether these structures were been drawn in the program or from internet sources

Answer: The source of the chemical structures were from internet source by PubChem. Thanks for your comment.

  1. There is no in vivostudy to support your work.

Answer: We are grateful for your observation, it is very important for this study, but For now we convert this work in a brief report to accomplish a recommendation of others reviwers, but we are working in this idea to a next article.

  1. Authors need to correct some grammatical mistakes

Answer: We corrected the grammatical mistakes, thank you for your observations.

7 ·    Please check the abbreviations throughout the manuscript

Answer: We checked the abbreviations as you reccomended, thank you for your observation.

  1. The discussion needs more improvement

Answer: We improved the discussion.

  1. The source of naringenin and miltefosine is not clarified

Answer: We clarified the source of naringenin and miltefosine on the manuscript as your recommendation. The sources of naringenin and other substances used in this work are commercial, acquired from Sigma aldrich. Miltefosine ara acquired from Cayman Chemical.

10 ·   The authors should cite more recent and updated references

Answer: We cited more recent and updated references on the manuscript.

We appreciate your questions and comments!!

Reviewer 2 Report

Comments and Suggestions for Authors

Dr. Vinícius Lessa et al provides a synergistic effects of naringenin and miltefosine against Leishmania amazonensis. The topic is very relevant, since Leishmaniasis have become a major health issue and a global problems. It will be more interesting if the following comment(s) should be assessed and taken necessary actions.

1.     the authors should give a figurative diagram that indicate mechanism of action (M/A) of the combination of naringenin and miltefosine against Leishmania amazonensis.

2.     If necessary for M/A, the authors may do some experiments regarding immunomodulatory properties of the combination because Miltefosine show immunomodulatory properties (Dorlo TP, PMID: 22833634). This combination should be enhanced immunomodulatory properties.

Author Response

  1. the authors should give a figurative diagram that indicate mechanism of action (M/A) of the combination of naringenin and miltefosine against Leishmania amazonensis.

Answer: We thank you for your observation, but to meet the suggestion of one of the reviewers to convert this article into a brief report, it will not be possible to add more experiments for mechanism of action for this work. But your suggestion it is very important to us and we are implement this idea in another work in continuity of this brief report.

  1. If necessary for M/A, the authors may do some experiments regarding immunomodulatory properties of the combination because Miltefosine show immunomodulatory properties (Dorlo TP, PMID: 22833634). This combination should be enhanced immunomodulatory properties.

Answer: We thank you for your observation, but to meet the suggestion of one of the reviewers to convert this article into a brief report, it will not be possible to add more experiments to this work.

We appreciate your questions and comments!!

Reviewer 3 Report

Comments and Suggestions for Authors

I have read the manuscript titled “In vitro Evaluation of the Combined Effects of Naringenin and Miltefosine on Leishmaniasis amazonensis.” There is increasing interest in enhancing the effectiveness to treat Leishmaniasis amazonensis by using miltefosine and naringenin. However, this manuscript can be published after it has been revised and the necessary questions have been addressed.
1. P2L61-65: This sentence "
Miltefosine (n-hexadecylphosphonocholine) a phospholipid that is 61 the hexadecyl monoester of phosphocholine with molecular weight of 407,6 g/mol, show 62 a lower toxicity when compared with the meglumine antimoniate, easy oral 63 administration, and efficiency rate in the elimination of the parasite close to that of first 64 and second choice drugs." was mentioned in manuscript. Miltefosine is known for its good efficacy and low toxicity. However, the author chose to add naringenin to the treatment. Could you please clarify the reason for including naringenin?
2. In this study, the authors conducted an in silico study, evaluated antipromastigote activity in vitro, and determined ROS levels in promastigotes. However, the manuscript provides extensive information on the toxicity profiles of drugs currently used to treat this disease. Could the authors provide information on the toxicity of using miltefosine in combination with naringenin?
3. 
To determine the ROS levels in promastigotes, the authors used doses of 12.5 μM of miltefosine and 114 μM of naringenin. Could the authors please explain why these specific doses were chosen?
4. Could the authors please specify the source of the Leishmania amazonensis strain used in the study, as well as the sources of naringenin, miltefosine, and any other substances used?
5. In this manuscript, the author intends to publish it as an article in this journal. However, in my opinion, the study is not yet comprehensive enough to qualify as a full article. The author might have additional or updated information to include.

Comments on the Quality of English Language

Minor editing of the English language is required.

Author Response

  1. P2L61-65: This sentence "Miltefosine (n-hexadecylphosphonocholine) a phospholipid that is 61 the hexadecyl monoester of phosphocholine with molecular weight of 407,6 g/mol, show 62 a lower toxicity when compared with the meglumine antimoniate, easy oral 63 administration, and efficiency rate in the elimination of the parasite close to that of first 64 and second choice drugs." was mentioned in manuscript. Miltefosine is known for its good efficacy and low toxicity. However, the author chose to add naringenin to the treatment. Could you please clarify the reason for including naringenin?

R-  We thanks you for your observation. We are serching for differents sources of leishmanicidal compounds that has a low toxicity in human cells in comparison the first line drugs used in treatment of leishmaniasis, and the strategy to use a low toxic natural compound in combination of a comercial drug for the optimization the treatment with miltefosine, despite this commericial drug had a good efficacy and low toxicity, there are some resistence isolates of Leishmania amazonesis, the porpuse of this investigation are to mitigate this problem too. Thank you for your observation.

  1. In this study, the authors conducted an in silico study, evaluated antipromastigote activity in vitro, and determined ROS levels in promastigotes. However, the manuscript provides extensive information on the toxicity profiles of drugs currently used to treat this disease. Could the authors provide information on the toxicity of using miltefosine in combination with naringenin?

R- We thanks you for your observation, but to meet your suggestion in item 5, we decide to convert this full article format into a brief report, it will not be possible to add more experiments.  But your suggestion it is very important to us and we are implement this idea in another work in continuity of this brief report.

  1. To determine the ROS levels in promastigotes, the authors used doses of 12.5 μM of miltefosine and 114 μM of naringenin. Could the authors please explain why these specific doses were chosen?

R- We appreciate your observation, these specific doses were selected because they were above the IC50 value of the compounds and for these experiments we needed a reasonable amount of parasites to perform the ROS test, facilitating the investigation using this technique and making it possible to observe some effect.

  1. Could the authors please specify the source of the Leishmania amazonensis strain used in the study, as well as the sources of naringenin, miltefosine, and any other substances used?

R- Yes, the sources of naringenin and other substances used in this work are commercial, acquired from Sigma aldrich. Miltefosine ara acquired from Cayman Chemical.

  1. In this manuscript, the author intends to publish it as an article in this journal. However, in my opinion, the study is not yet comprehensive enough to qualify as a full article. The author might have additional or updated information to include.

R- We appreciate your observation, and give us the alternative to submit this article in the format of brief report to this journal.

We appreciate your questions and comments!!

Reviewer 4 Report

Comments and Suggestions for Authors

The article was submitted for review - In vitro evaluation of the combinatorial effect of naringenin 2 and miltefosine against Leishmania amazonensis.

The article is relevant and of interest to a practicing physician.

Leishmaniasis is a disease caused by single-celled protozoa of the genus Leishmania. Among the clinical forms caused by L. amazonensis, diffuse cutaneous leishmaniasis (DCL) is difficult to treat with conventional drugs; The main characteristic of this clinical form is a Th2 response with expression of interleukin-10, IL-4 and low expression of interferon gamma (IFN), leading to anergy of the patient towards the parasite, high parasite loads and deforming lesions causing physical stigmas and psychological effects. Efforts have been directed toward plant-based treatments and their metabolites, particularly flavonoids, which have cancer preventive, antioxidant, and leishmanicidal properties. The authors chose the flavonoid naringenin (5,7-dihydroxy-2-(4-hydroxyphenyl)chromman-4-one). In addition to its important immunomodulatory properties, naringerin has demonstrated potent in vitro activity against Leishmania donovani promastigotes and amastigotes.

The work used modern laboratory research methods. Anti-leishmanial dose/response activity, isobologram, calculation of dose reduction index, fractional inhibitory concentration index and reactive oxygen species tests were performed on in vitro platforms using reference promastigote forms of L. amazonensis (IFLA/BR/67/PH8) to evaluate the combinatorial effect of naringenin and Miltefosine. Adequate statistical processing of the data was carried out. The study demonstrated the in vitro viability of the combined action of the flavonoid with miltefosine treatment, opening the space for further research into the relationship of natural compounds with drugs used to treat L. Amazonensis. The bibliography contains 33 literary sources, including recent ones.

The article may be recommended for publication without changes.

Author Response

We appreciate your questions and comments!! Thank you for your observations for this work, they are very important.

Round 2

Reviewer 1 Report

Comments and Suggestions for Authors

·       The authors said ‘we convert this work in a brief report to accomplish a recommendation of others reviwers

But till now, the article type in the title of the revised manuscript is  Article, not brief report.

·       The structures were from internet sources not drawn in any program as ChemDraw.

·       The references are still not updated.

·       Grammatical, alignment, and typographical errors are noted in the manuscript and the author’s response.

 I think that this manuscript does not reach the level of a strong journal such as Pharmaceuticals

Comments on the Quality of English Language

·       Grammatical, alignment, and typographical errors are noted in the manuscript and the author’s response.

Author Response

Comments 1: 

The authors said ‘we convert this work in a brief report to accomplish a recommendation of others reviwers

But till now, the article type in the title of the revised manuscript is  Article, not brief report.

Response 1:  We agree with this comment and we modified the article type in the title to brief report. Thank you for pointing this out.

Comment 2: The structures were from internet sources not drawn in any program as ChemDraw.

Response 2: Thank you for point this out. We don't drawn in any program because the structures are well described in puchem and we used this source because we based in another work of our team " In vitro anti-leishmania activity of triclabendazole and its synergic effect with amphotericin B by Borges, Beatriz Santana et al.

Comments 3: The references are still not updated.

Response 3:  Thank you for your sugestion. We re-evaluated the references and they are appropriate to the text. Thank you for point this out.

Comment 4:  Grammatical, alignment, and typographical errors are noted in the manuscript and the author’s response.

Response 4: Thank you for your observations. The manuscript were revised for more corrections.

Reviewer 3 Report

Comments and Suggestions for Authors

The author has satisfactorily responded to the questions and followed the recommendations I provided. I believe this manuscript has been adequately revised and is suitable for publication in Pharmaceuticals.

Author Response

Comment 1: The author has satisfactorily responded to the questions and followed the recommendations I provided. I believe this manuscript has been adequately revised and is suitable for publication in Pharmaceuticals.

Response: We appreciate your questions and comments!! Thank you for your observations for this work and thank you for your recommendation to switch the type of article, your comments were very important for us.